# Acetyl Tributyl Citrate Exposure at Seemingly Safe Concentrations Induces Adverse Effects in Different Genders of Type 2 Diabetes Mice, Especially Brain Tissue

**DOI:** 10.3390/toxics11100877

**Published:** 2023-10-23

**Authors:** Yuchao Zhang, Zhihuai Zhang, Sijie Zhu, Liangyu Liu, Xudong Liu, Xu Yang

**Affiliations:** 1Department of Brewing Engineering, Moutai Institute, Renhuai 564507, China; zyc10271027@163.com; 2Department of Food Science and Engineering, Moutai Institute, Renhuai 564507, China; dearrambo@163.com (Z.Z.); 18798006468@163.com (S.Z.); aleliuliangyu@163.com (L.L.); 3College of Food Science and Engineering, Ocean University of China, Qingdao 266003, China; 4College of Basic Medical Science, Hubei University of Science and Technology, Xianning 437100, China

**Keywords:** acetyl tributyl citrate, type 2 diabetes, brain tissue damage, gender-dependent, oxidative stress, inflammation, glial cell homeostatic level

## Abstract

Acetyl tributyl citrate (ATBC) is a widely used phthalate substitute. Although ATBC is considered to be with a safe dosage of up to 1000 mg/kg/day, studies on its effects in some sensitive populations, such as diabetic patients, are relatively rare. Epidemiological studies have shown that there is a link between diabetes and nervous system diseases. However, toxicological studies have not fully confirmed this yet. In this study, glycolipid metabolism, cognitive deficits, brain tissue damage, levels of neurotransmitters, beta-amyloid plaques (Aβ), hyperphosphorylated tau protein (p-Tau), oxidative stress and inflammation, as well as glial cell homeostatic levels in the brain tissue of type 2 diabetes (T2DM) mice, were determined after ATBC exposure (0, 2, 20, and 200 mg/kg/day) for 90 days. The results confirmed that ATBC exposure aggravated the disorder of glycolipid metabolism and caused cognitive deficits in T2DM mice; induced histopathological alterations and Aβ and p-Tau accumulation, and reduced the levels of 5-hydroxytryptamine and acetylcholine in T2DM mouse brains; oxidative stress and glial cell homeostatic levels in T2DM mouse brains were also changed. Some of the adverse effects were gender-dependent. These findings support the theory that T2DM mice, especially males, are more sensitive to ATBC exposure. Although the safe dose of ATBC is high, prolonged exposure at seemingly safe concentrations has the potential to aggravate diabetes symptoms and cause brain tissue damage in T2DM mice.

## 1. Introduction

Due to its serious threat to human health, diabetes is considered to be a worldwide public health problem that represents a significant societal burden. It is estimated that the number of people with diabetes worldwide will reach 630 million by the year 2050 [1]. As a typical endocrine system disease, diabetes is a glucose metabolism disorder caused by a lack of insulin or insulin resistance. It is categorized into type 1 diabetes (absolute lack of insulin) and type 2 diabetes (T2DM, insulin resistance or relative lack of insulin). About 90% of all patients have T2DM. If not effectively controlled, diabetes can lead to many complications, especially nervous system and cardiovascular complications [2,3]. It is the cause of 4 million premature deaths each year worldwide [1].

T2DM patients are likely to develop comorbidities such as nervous system diseases, including Alzheimer’s disease (AD) [4,5,6]. The probability of experiencing T2DM complicated by AD has reached 30%, and AD has been referred to as “type 3 diabetes” [7]. T2DM complicated with nervous system diseases undoubtedly introduces a challenge to its treatment. Although some specific genes related to diabetes and nervous system diseases have been identified, genetic factors fail to fully account for their pathogenesis [8,9]. Therefore, the environmental and lifestyle roles in the occurrence of diseases are being actively probed.

Plasticizers are usually used to impart flexibility and durability to plastic products. Worldwide, more than 6 million tons of plasticizers are produced and consumed each year [10]. As the most widely used plasticizers, phthalates are found in many consumer products, such as beauty products, oral medications, medical devices, food and beverage containers, and children’s care products and toys. Humans are exposed to phthalates via multiple routes [11]. However, phthalates have been restricted in many countries and regions due to their endocrine-disrupting effects [12].

Acetyl tributyl citrate (ATBC) is a Food and Drug Administration-approved food contact chemical that has been proposed as an alternative to phthalates [13]. Like other phthalates, ATBC has been shown to migrate from food packaging material and medical equipment into the products that they contain even more rapidly than the potent endocrine disruptor di-2-ethylhexyl phthalate [14,15]. Various studies have employed dosages in the range of 50–1000 mg/kg/day to examine the systemic, reproductive, and developmental toxicity of ATBC in animal models [6,13]. Based on these studies, ATBC was determined to be a safe alternative to phthalates. ATBC is considered to be a high-safety plasticizer (the safe dosage is up to 1000 mg/kg/day); our previous studies have demonstrated that at 20 mg/kg/day ATBC exposure does not cause the functional impairment of brain tissue in normal mice [16,17]. However, studies on its effects on some sensitive populations, such as diabetic patients, have not received much attention. Since diabetes is a disease prone to complications, it is important to study whether ATBC exposure can promote diabetes complicated with nervous system diseases.

The present study examined the effects of ATBC exposure (particular in concentrations that are relatively safe for normal mice) on T2DM mice, especially in the brain tissue. The key upstream events for the results were evaluated to explore the possible mechanism of the effects on T2DM mice brain tissue. These findings provide a reference for understanding diabetes complicated by nervous system diseases.

## 2. Materials and Methods

All experimental procedures were approved by the Office of Scientific Research Management of Moutai Institute (Renhuai, China). Animal experiments were conducted according to the National Institutes of Health Guide for the Care and Use of Laboratory Animals with a certificate of Application for the Use of Animals dated 5 March 2019 (approval ID: MI-IACUC-2019-003).

### 2.1. Animals

SPF C57BL/6 mice (six weeks old, 22 g) were maintained in pathogen-free cages at 20–25 °C with 50–70% humidity and a 12 h light/dark cycle. The animals were provided with commercially available chow (normal or high-fat diet) and filtered water ad libitum.

### 2.2. Main Reagents and Kits

Normal-diet (ND, D10001) and high-fat-diet (FD, D12492) chow was purchased from Research Diets Inc., New Brunswick, NJ, USA. Streptozocin (STZ), 2′,7′-dichlorofluorescin diacetate (DCFH-DA), 2-thiobarbituric acid (TBA), and 3-carboxy-4-nitrophenyl disulfide (DTNB) were purchased from Sigma-Aldrich Inc., St. Louis, MO, USA. Blood glucose test paper and a meter were purchased from Johnson & Johnson Inc., New Brunswick, NJ, USA. Insulin, triglyceride (TG, A110-1-1), cholesterol (TC, A110-1-1), acetylcholine (ACh, A105-1-1), 5-hydroxytryptamine (5-HT, H104-1-1), and superoxide dismutase (SOD, A001-3-2) activity kits, 4-hydroxynonenal (4-HNE, H268-1-2) were purchased from Nanjing Jiancheng Bioengineering Institute, Nanjing, China. Aβ_1–42_ ELISA kit (KMB3441) and anti-Tau (13-6400) and pTau231 antibodies (701056) were purchased from Invitrogen Inc., Carlsbad, CA, USA. In addition, 8-hydroxy-2-deoxyguanosine (8-OHdG, 4380-192-K) ELISA kits were purchased from R&D Systems Inc., Minneapolis, MN, USA. Anti- IBA1 (ab178846) and GFAP (ab7260) antibodies and cytokine arrays (ab133993) were purchased from Abcam Inc., Shanghai, China.

### 2.3. Establishment of Mouse T2DM Model

Mice were acclimatized to ND chow for one week. Then, 20 mice (10 males and 10 females) were randomly selected for ND throughout the whole experimental period, while the remaining mice were fed FD chow for four weeks and then injected with STZ (50 mg/kg) intraperitoneally. These mice were then fasted and deprived of water for 12 h before the next injection. Next, the mice were injected with STZ once every three days. As a result, the mice received a total of three STZ injections during the whole experimental period. Except for 12 h fasting before each injection, the mice were free to eat and drink water at other times. One week after the completion of three STZ injections, the mice were fasted for 12 h and their blood samples were collected from the tail vein to measure their fasting blood glucose (FBG) levels. The model was considered successful when the FBG level was >11.1 mol/L [18,19]. Then, 40 male and 40 female T2DM mice were randomly selected. Finally, 20 healthy mice (10 males and 10 females) and 80 T2DM mice (40 males and 40 females) were obtained for subsequent experiments.

### 2.4. ATBC Exposure Protocol

A total of 10 healthy male mice and 10 healthy female mice were used as the two control groups (CK). Based on a prior study on the migration of ATBC in food and another in vivo neurotoxicity study on other plasticizers [14,20], the 40 T2DM male mice were divided into four groups that were, respectively, exposed to 0, 2, 20, and 200 mg/kg/day of ATBC via intragastric route for 90 consecutive days. The group of 40 T2DM female mice was consistent with that of T2DM male mice. The mouse grouping and treatment are shown in Table 1.

### 2.5. Increase in Body Weight, Food Consumed, and Water Intake

The body weight, food consumed and water intake of mice were recorded every day throughout the entire experimental period. The increase in body weight for the 90-day period were calculated as follows:Body weight increase (g) = body weight (90th day) − body weight (0th day)

### 2.6. Determination of FBG Level and Glucose Tolerance

The mice were fasted without water deprivation for 12 h after the last ATBC exposure. Then, blood samples were collected via tail cuts. The FBG levels were determined using a glucometer. Next, all of the mice were gavaged with a glucose solution at a dose of 2 g/kg (body weight). The blood samples were again collected via tail cuts 0, 0.5, 1, and 2 h after glucose gavage to determine blood glucose levels. The area under the curve (AUC) for blood glucose results was also calculated [21].

### 2.7. TC, TG, LDL-C, HDL-C, and Insulin Content Assessment

Mouse TC, TG, LDL-C, HDL-C, and insulin content detection kits were used to determine TC, TG, LDL-C, HDL-C, and insulin levels in mouse blood following instructions provided by the manufacturer.

### 2.8. Morris Water Maze (MWM)

The water maze experiment was used to determine the cognitive ability changes of T2DM mice after ATBC exposure as previously described [22]. Briefly, the swimming training continued from 84th day to 88th day. The escape latency (the time it takes mice to find the platform) were detected. On the 89th day, no MWM was performed; on the 90th day the MWM was performed again, but no platform in the pool. The swimming time in the southwest (SW) quadrant (quadrant where the platform is located) and the swimming route were recorded. Image-Pro Plus 6.0 (Bethesda, MD, USA) was used to analyze the average optical density (OD) of the swimming route heat map in the SW quadrant.

### 2.9. Liver, Pancreas, and Brain Sample Histological Examination

Liver, pancreas, and brain samples were collected 24 h after the last ATBC exposure (the mice were sacrificed by cervical dislocation) and fixed in a mixture of saturated 2,4,6-trinitrophenol/formalin/glacial acetic acid [15:5:1 *v*/*v*/*v*] for 24 h at room temperature. Samples were then embedded in paraffin and sectioned into 10 μm slices for hematoxylin and eosin (H&E) or Nissl staining [23]. The vacillations of liver and pyramidal cell, the area of the islets and the OD of Nissl staining were determined using Image-Pro Plus 6.0.

### 2.10. Tissue Sample Preparation

The collected brains were weighed on a completely automatic electronic balance. Tissue samples were placed in 10 mL/g of ice-cold 1 × phosphate-buffered saline (pH = 7.5) and homogenized using a glass homogenizer. Homogenates were then centrifuged at 10,000 rpm for 10 min at 4 °C. The supernatant was collected and kept at −70 °C until needed for further assays. The supernatant’s protein concentration was determined using the Lowry assay [23].

### 2.11. ACh, 5-HT, and Aβ_1–42_ Assessment

Mouse 5-HT, ACh, and Aβ_1–42_ content detection kits were used to detect 5-HT, ACh, and Aβ_1–42_ in mouse brain tissue following instructions provided by the manufacturer.

### 2.12. Western Blot Analysis of Tau Phosphorylation

Western blot analysis was carried out as previously described with some modifications [24]. Briefly, 5 μL samples of brain extracts were diluted 1:2 in the sample buffer and then denatured by boiling at 100 °C for 5 min. SDS-PAGE was used to separate proteins, followed by transfer onto polyvinylidene difluoride membranes at 30 V for 1 h. Membranes were probed with antibodies specific for total Tau (TAU) or site-specific Tau phosphorylation (pTAU231).

### 2.13. Assessment of Reactive Oxygen Species (ROS), Malondialdehyde (MDA), and Glutathione (GSH)

The ROS, MDA, and GSH content in the mouse brains was measured using DCFH-DA, TBA, and DTNB, respectively, as previously described [25].

### 2.14. Analysis of 4-HNE and 8-OHdG Content, and SOD Activity

ELISA kits were used to measure 4-HNE and 8-OHdG content. SOD activity was determined using an SOD activity kit. The kits were used following instructions provided by the manufacturer.

### 2.15. Cytokine Array

A total of 1 mg of protein from mouse brain tissue was loaded onto the antibody-coated cytokine array nitrocellulose membranes. The membranes were incubated with secondary antibodies and the cytokines were visualized according to the manufacturer’s protocol.

### 2.16. Immunofluorescence (IF) Analysis of GFAP and IBA1

Immunofluorescence analysis was performed as previously described with some modifications [26]. Briefly, the brain sections were incubated overnight at room temperature with mouse antibodies against GFAP and IBA1. Subsequently, brain sections were washed and incubated with FITC-conjugated secondary antibody at 37 °C for 60 min, washed again, mounted, and examined. The OD of GFAP and IBA1 fluorescence were determined using Image-Pro Plus 6.0.

### 2.17. Statistical Analyses

All collected data were analyzed using one-way ANOVA followed by a Tukey’s test. Values of *p* < 0.05 were considered statistically significant. Data analyses were carried out using SPSS 13.0 software (Chicago, IL, USA). Statistical analysis graphs were generated using GraphPad Prism 8.0 (San Diego, CA, USA).

## 3. Results

### 3.1. Glucose and Lipid Metabolism Index Changes in Mice after ATBC Exposure

The body weight increases in T2DM and T2DM + ATBC_(Low)_ mice were significantly lower than those in CK mice after 90 days of ATBC exposure (Figure 1a). The body weight decreased and male mice lost significantly more body weight than female mice after exposure to medium and high concentrations of ATBC. The T2DM mice demonstrated a higher level of food consumed and water intake than CK mice (Figure 1b,c). Although the ATBC exposure at a low concentration did not significantly affect the food consumed and water intake of T2DM mice, medium and high concentrations of ATBC further improved the food and water intake of T2DM mice. This increase was more significant in male mice.

The FBG levels of T2DM and T2DM + ATBC_(Low)_ mice were significantly higher than those of the CK mice after 90 days of ATBC exposure, and both exceeded the modeling standard; there was no significant difference between T2DM and T2DM + ATBC_(Low)_ mice. Meanwhile, the FBG levels were further significantly improved after exposure to ATBC at medium and high concentrations compared with T2DM and T2DM + ATBC_(Low)_ mice. And there was also a difference between male and female mice: the levels in male mice increased more significantly (Figure 1d).

The AUC for each group of mice showed the same trend as that for FBG levels (Figure 1e). The insulin levels in T2DM mice were decreased compared to the CK group. They decreased even further after exposure to medium and high concentrations of ATBC. The decreases in male mice were more pronounced, such that there was a significant difference between the male and female mice (Figure 1f).

The TC, TG, and LDL-C levels of T2DM mice were significantly higher than those of CK mice, and exposure to low and medium concentration of ATBC had no effects (Figure 1g–i). However, the TC, TG, and LDL-C levels of male T2DM mice increased significantly after exposure to high concentrations of ATBC, while the female T2DM mice showed no changes in TG and LDL-C levels. The trend in HDL-C change was reversed (Figure 1j). The HDL-C levels in T2DM and T2DM + ATBC_(Low)_ mice were significantly lower than those in CK mice. They decreased even further after exposure to medium and high concentrations of ATBC. The levels in male mice decreased more, such that there was a significant difference between male and female mice.

### 3.2. Cognitive Ability Changes after ATBC Exposure

According to Figure 2a, after 5 days swimming training, on the 88th day, the escape latency of T2DM mice had not changed. However, after medium and high concentrations of ATBC exposure, the escape latency of mice was increased significantly, while after the forgetting period (the 89th day), on the 90th day, the time of T2DM + ATBC_(Mid)_ and T2DM + ATBC_(High)_ mice stayed in the platform quadrant was remarkably shorter than other groups (Figure 2b). Through the heat map of mouse swimming routes on the 90th day (Figure 2c), the relatively concentrated areas (red areas) of each group mice in SW quadrant were different. After medium and high concentrations of ATBC exposure, the concentrated routes of T2DM mice in the platform quadrant gradually decreased (Figure 2d). And the gender-based differences were observed after medium and high concentrations of ATBC exposure.

### 3.3. Histological Analysis of Liver and Pancreas after ATBC Exposure

The liver cells of mice in the CK group exhibited normal morphology, with no fatty lesions, obvious swelling, or atrophy; however, the structural abnormalities of hepatocytes became more obvious when the ATBC exposure concentration increased (Figure 3a). As shown in Figure 3c, the liver cells of the T2DM and T2DM + ATBC_(Low)_ groups showed a sparse cytoplasm with vacuoles. And the vacuolation of T2DM mice further increased after medium and high concentrations of ATBC exposure, while the male mouse hepatocytes in the T2DM + ATBC_(Mid)_ and T2DM + ATBC_(High)_ group showed large areas of cytoplasmic sparsity and vacuolation.

The pancreas islets and peripheral cells of mice in the CK group were regular and complete (Figure 3b). The islets were round or oval, the islet edges were clear, the islet size was normal, and the cells in islets were evenly distributed. Islets in the T2DM group showed irregular morphology and irregular edges. Islet irregularity and atrophy in T2DM mice became more pronounced after ATBC exposure, and the damage gradually increased with the increase in exposure concentration. After exposure to ATBC at a high concentration, the islets of female and male T2DM mice were still visible, but the edges and boundaries of the islets were unclear, and the morphological structure was changed. As shown in Figure 3d, the total areas of the islets of the T2DM and T2DM + ATBC_(Low)_ groups were decreased, and the areas of the T2DM mice further reduced after medium and high concentrations of ATBC exposure, while the male mouse in the T2DM + ATBC_(Mid)_ and T2DM + ATBC_(High)_ group showed more reduced total islet areas.

### 3.4. Histological Analysis of Hippocampus after ATBC Exposure

H&E staining results showed that pyramidal cells in the hippocampus region of mice from the CK, T2DM, and T2DM + ATBC_(Low)_ groups were arranged in an organized manner and had a clear polygonal shape (Figure 4a). The apical dendrites of each cell were clearly visible. However, after ATBC exposure at medium and high concentrations, the arrangement of T2DM mice pyramidal cells became loose and disordered, with swelling deformations and cytoplasmic vacuolation, shortening, or even apical dendrite disappearance. As shown in Figure 4c, the vacuolation of T2DM mice pyramidal cells increased after medium and high concentrations of ATBC exposure.

Nissl staining of the hippocampus is shown in Figure 4b. The CK, T2DM, and T2DM + ATBC_(Low)_ groups showed the clear cytoplasmic staining of cells, but after ATBC exposure at medium and high concentrations, the cytoplasmic staining of pyramidal cells became lighter. As shown in Figure 4b, decreases in the OD of Nissl staining were observed in mice from the T2DM + ATBC_(Mid)_ and T2DM + ATBC_(High)_ groups.

### 3.5. ACh, 5-HT, Aβ_1–42_, and Tau Phosphorylation Level Changes in Mouse Brains after ATBC Exposure

The ACh levels in T2DM mice were significantly decreased after exposure to medium and high concentrations of ATBC (Figure 5a). Gender-based differences were also observed. The decreases in female mice were lower than those in the male mice. After exposure to ATBC at high concentration, the 5-HT level changes in T2DM mouse brains showed the same trend as those for ACh (Figure 5b). The Aβ_1–42_ expression in T2DM male mice significantly increased compared to that in the CK group. It increased even further after exposure to medium and high concentrations of ATBC in T2DM male mice (Figure 5c). However, the Aβ_1–42_ expression showed an increase in T2DM female mice only after exposure to high concentrations of ATBC. The relative expression of pTAU231 increased significantly after ATBC exposure, and gender-based differences were also observed (Figure 5d).

### 3.6. Oxidative-Stress Level Changes in Mouse Brains after ATBC Exposure

A significant increase in ROS levels was observed in T2DM mouse brains. The ROS levels increased further after exposure to medium and high concentrations of ATBC (Figure 6a). However, the increase was less pronounced in female T2DM mice than in male mice. The MDA and 4-HNE levels in T2DM mouse brains showed the same trend as those for ROS (Figure 6b,c). The GSH levels decreased significantly in T2DM mouse brains. They decreased even further after exposure to medium and high concentrations of ATBC and showed gender-based differences in the T2DM + ATBC_(Mid)_ group (Figure 6d). The SOD activity in T2DM mouse brains decreased significantly after exposure to high concentrations of ATBC. Gender-based differences were also evident (Figure 6e). The 8-OHdG level changes in T2DM mouse brains are shown in Figure 6f. The 8-OHdG levels in male T2DM mice increased significantly after exposure to high concentrations of ATBC, while the female T2DM mice showed no changes.

### 3.7. Cytokine Level Changes in Mouse Brains after ATBC Exposure

Significant increases in IL-2, IL-17, INF-γ, sTNFRI, and TNF-α levels were observed in the T2DM mouse brains. The levels increased further after exposure to medium and high concentrations of ATBC (Figure 7). These increases showed gender-based differences in the T2DM + ATBC_(Mid)_ and T2DM + ATBC_(High)_ groups. MCP-1, IL-6, and MCP-5 levels in the T2DM mouse brains were increased after exposure to medium or high concentrations of ATBC. Gender-based differences in the T2DM + ATBC_(Mid)_ and T2DM + ATBC_(High)_ groups were also demonstrated.

### 3.8. Glial Activation in Mouse Brains after ATBC Exposure

In response to ATBC espouse, glial cells became more intense after exposure to medium and high concentrations of ATBC. And astrocytes were characterized by an increased GFAP-positive cell body volume and more ramified branches (Figure 8a). Significant increases in GFAP expression levels were observed in the T2DM- and T2DM + ATBC_(Low)_-group mice compared to the CK-group mice. And the expression level increased gradually after exposure to medium or high concentrations of ATBC (Figure 8c). They experienced IBA1-positive cell size increases, neurite retraction and coarsening, and spines appearing on the neurites (Figure 8b). Similar to GFAP expression, significant increases in IBA1 expression levels in the T2DM- and ATBC-exposure-group mice were observed (Figure 8d). And the male mouse brain in the T2DM + ATBC_(Mid)_ and T2DM + ATBC_(High)_ group showed more increased GFAP and IBA1 expression.

## 4. Discussion

Diabetes is a high-prevalence disease that greatly threatens patients’ lives and poses a heavy burden on their families and society in general, which is mostly attributable to the complications of this disease, especially concurrent neurological and cardiovascular disorders. Although some susceptibility genes for diabetes have been identified, they are not fully responsible for the pathogenesis of diabetes and diabetes complications. Environmental factor exposure is believed to be an important consideration for the incidence of various diseases and is highly concerning. The use of traditional plasticizers has been strictly controlled due to their multiple toxicities. ATBC is a new low-toxicity plasticizer that has been widely used in food, pharmaceuticals, and some objects of daily necessities [27]. The European Food Safety Authority reported that the No-Observed-Adverse-Effect-Level of ATBC was 100 mg/kg body weight (BW)/day. Recent studies have reported the estimated the daily intake of ATBC via dust ingestion as 89.9 and 1270 ng/kg BW/day for adults and infants, respectively [28]. According to the FDA report, some patients may orally take up to 20.2 mg of ATBC per day, due to its usage in tablets and capsules [29]. In the present study, ATBC exposure effects on T2DM mice, especially their brain tissue, at safe dosages were investigated to provide a reference for the better understanding of diabetes when it is complicated with nervous system diseases.

In the present study, a T2DM mouse model was successfully constructed by feeding mice a high-fat diet and administering them multiple injections of low-dose STZ. The T2DM mice showed typical symptoms of diabetes, increased food and water intake, decreased weight gain, abnormal FBG and glucose tolerance, and decreased insulin levels. Some lipid metabolism indices showed that T2DM mice also had abnormal lipid metabolism. Some studies have previously shown that diabetes is often accompanied by abnormal lipid metabolism [30,31]. Low-concentration ATBC exposure did not significantly increase the abnormal glucose level and lipid metabolism of T2DM mice, while exposure to medium and high concentrations of ATBC aggravated them and showed gender-based difference. Male T2DM mice were more sensitive to medium and high concentrations of ATBC. The histopathological observations of mouse liver and pancreas also showed that ATBC exposure at medium and high concentrations significantly increased liver and pancreas tissue lesions in T2DM mice. These results indicated that medium and high concentrations of ATBC significantly aggravated glucose- and lipid-metabolism disorder in T2DM mice.

Studies have shown that abnormal glycolipid metabolism is considered to be one of the causes of neurological diseases [32,33,34]. The present study focused on the effect of ATBC exposure in the brain tissues of T2DM mice. The MWM test was selected to test the cognitive ability of each group of mice; high-concentration ATBC exposure could increase the escape latency and decrease the platform-quadrant retention time. The results showed that medium- and high-concentration ATBC exposure induced cognitive deficits in T2DM mice, while these changes had a gender difference. Histopathological observations of the hippocampal region revealed that pyramidal neurons in T2DM mice showed significant morphological and structural changes after exposure to medium and high concentrations of ATBC, while the Nissl substance content decreased. As a key component of neurotransmitter synthesis, the absence of the Nissl substance affects neurotransmitter synthesis [35]. The decrease in 5-HT and ACh levels in brain tissues of T2DM mice exposed to medium and high concentrations of ATBC may be related to the absence of the Nissl substance. These changes may lead to abnormal brain function. The present study also found that ATBC exposure increased the accumulation of Aβ_1–42_ and p-Tau protein in T2DM mouse brains, which are considered to be two indicators for the evaluation of AD. Abnormal glucose and lipid metabolism has also been confirmed by the studies on the pathogenesis of AD [7]. These results have also shown gender differences, especially after exposure to medium and high concentrations of ATBC. The present studies on diabetes and its complications have also revealed gender differences in the incidence of diabetes and related complications [36,37]. This suggests that exposure to medium and high concentrations of ATBC may lead to complications of nerve injury while aggravating the disorder of glycolipid metabolism in T2DM mice.

Studies have demonstrated that oxidative stress is one of the essential mechanisms for the pathogenesis of diabetes [38,39]. Brain tissues are more susceptible to oxidative stress due to their high oxygen consumption, high lipid content, and low antioxidant capacity [40]. The studies on gender differences in oxidative stress have shown that females may exhibit greater antioxidant capacity. These differences may be caused by differences in sex chromosomes, the expression of sex-specific genes on autosomes, sex hormones, and their effects on the organs and systems [41,42]. In the present study, the oxidative stress level changes after ATBC exposure were determined. The results showed that ROS and lipid peroxidation levels (MDA, 4-HNE) increased and GSH levels decreased, while SOD activity and 8-OHdG levels did not change in the brain tissue of T2DM mice. However, exposure to medium and high concentrations of ATBC aggravated the ROS levels. Furthermore, lipid peroxidation increased and GSH levels decreased. In addition, SOD activity decreased significantly, while 8-OHdG levels increased. Gender-based differences were also evident. These results indicate that the levels of oxidative stress in the brain tissues of T2DM mice of different genders were differentially increased after exposure to medium and high concentrations of ATBC.

Since glial cells are the innate immune cells in the brain tissue, abnormal homeostatic levels of these have been shown to be associated with neurological diseases and nerve damage. Responding to toxic exposure, glial cells could induce the production of ROS and cytokines, which damage nerve cells and stimulate glial cell activation, leading to injury and disease [43]. In the present study, glial cell homeostatic level in the brain tissue of T2DM mice was changed, which was further enhanced when the concentrations of ATBC increased. The results showed that the levels of some cytokines increased in the brain tissue of T2DM mice, while the expression levels of cytokines were further significantly increased after exposure to medium and high concentrations of ATBC. Most of these cytokines are pro-inflammatory factors. There were also gender-based differences in the cytokine content changes. The results suggested that medium and high concentrations of ATBC induced high levels of oxidative stress, and the glial cell homeostatic level changed and enhanced the over-expression of pro-inflammatory factors, leading to the exacerbation of neuroinflammation and brain tissue injury.

## 5. Conclusions

In the present study, the effects of ATBC exposure on T2DM mice, especially on brain tissue, were investigated using a range of seemingly safe concentrations. The results showed that when the exposure concentration reached 20 mg/kg/day, ATBC exposure exacerbated the disorder of glucose and lipid metabolism in T2DM mice and induced adverse effects on the T2DM mouse brain. However, these exposure concentrations have been shown to be safe for normal mice. This indicates that the seemingly safe ATBC exposure-induced brain tissue damage in T2DM mice does not exist independently. Although the ATBC exposure concentration used in this study was higher than the current environmental concentration, considering the high use of ATBC, high migration and the difference in tolerance between animals and humans, the exposure concentrations used in this study on mice has reference significance for future human exposure. They also suggest that oxidative stress and glial activation may be involved in the toxicity mechanisms. Furthermore, it was found that the brain tissue of male T2DM mice may be more sensitive to ATBC exposure. The present study indicated that ATBC exposure in T2DM mice at a dose that is safe for the normal population can induce significant brain tissue damage and promote the development of neurological complications, especially in males (Figure 9). In future studies, we plan to explore whether ATBC induces brain damage in mice through itself or metabolites, and how to induce oxidative damage and neuroinflammation; for example, we may focus on the effect of ATBC on the PPAR gamma receptor.

## Figures and Tables

**Figure 1 toxics-11-00877-f001:**
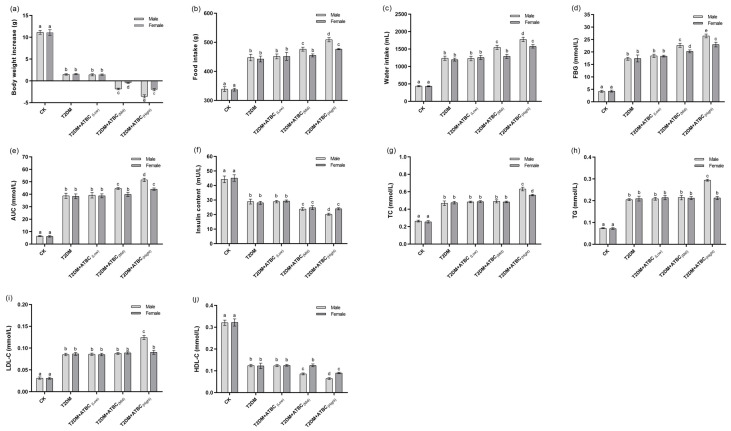
Glucose and lipid metabolism index changes in mice after ATBC exposure. (**a**) Body weight increase; (**b**) food consumed; (**c**) water intake; (**d**) FBG; (**e**) AUC; (**f**) insulin; (**g**) TC; (**h**) TG; (**i**) LDL-C; and (**j**) HDL-C. Different lowercase letters indicate significant differences (*p* < 0.05) between groups in the plots.

**Figure 2 toxics-11-00877-f002:**
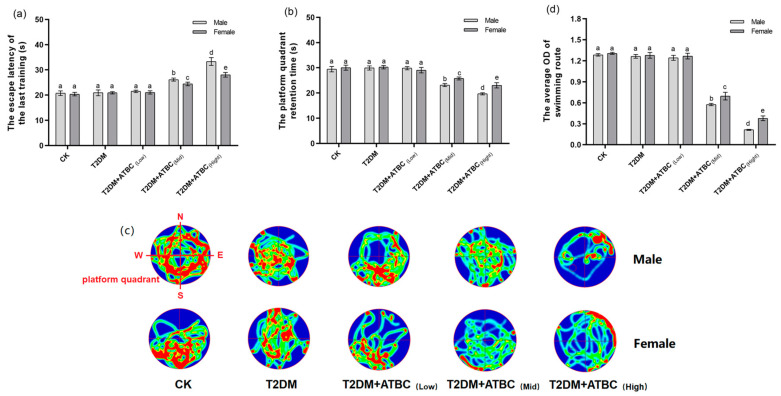
MWM test after ATBC exposure. (**a**) The escape latency of last training. (**b**) The platform quadrant retention time. (**c**) The swimming route (Green to red indicates the degree of concentration of the route). (**d**) The average OD (red) of the swimming route in the platform quadrant. Different lowercase letters indicate significant differences (*p* < 0.05) between groups in the plots.

**Figure 3 toxics-11-00877-f003:**
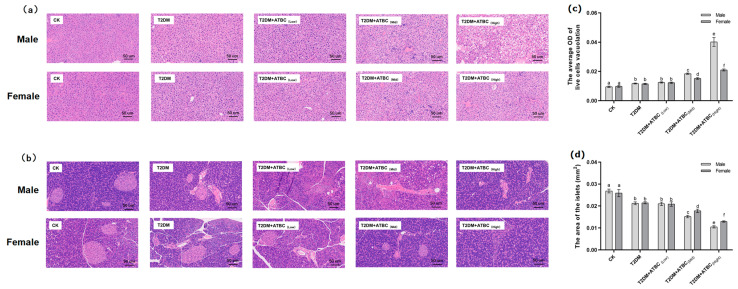
Histological analysis of liver and pancreas after ATBC exposure. (**a**) Liver (The whit part indicates sparse cytoplasm with vacuoles); (**b**) pancreas; (**c**) the average OD (white) of live cell vacuolation; and (**d**) the total area of the islets. (bar = 50 μm). Different lowercase letters indicate significant differences (*p* < 0.05) between groups in the plots.

**Figure 4 toxics-11-00877-f004:**
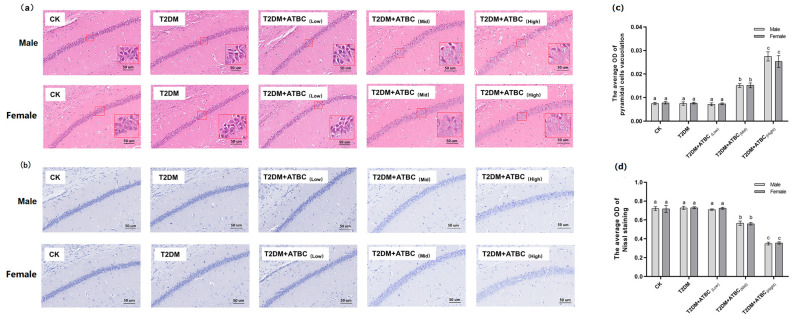
Histological analysis of hippocampus after ATBC exposure. (**a**) H&E staining (White part indicate cytoplasmic vacuolation); (**b**) Nissl staining (Blue part is Nissl substance); (**c**) the average OD of pyramidal cell vacuolation; (**d**) the average OD of Nissl staining. (bar = 50 μm). Different lowercase letters indicate significant differences (*p* < 0.05) between groups in the plots.

**Figure 5 toxics-11-00877-f005:**
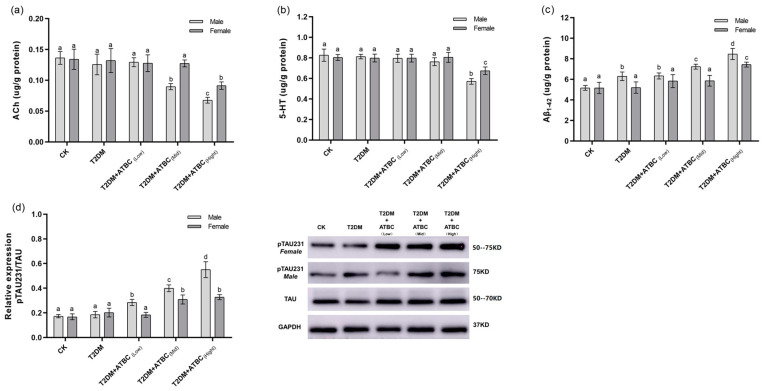
ACh, 5-HT, Aβ_1–42_, and Tau phosphorylation level changes in mouse brains after ATBC exposure. (**a**) ACh; (**b**) 5-HT; (**c**) Aβ_1–42_; and (**d**) Tau phosphorylation (TAU: Total Tau protein, GAPDH: intrinsic reference protein). Different lowercase letters indicate significant differences (*p* < 0.05) between groups in the plots.

**Figure 6 toxics-11-00877-f006:**
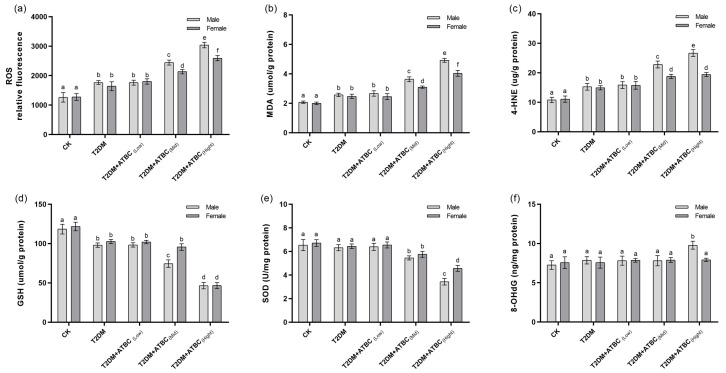
Oxidative-stress level changes in mouse brains after ATBC exposure. (**a**) Relative ROS fluorescence; (**b**) MDA concentration; (**c**) 4-HNE concentration; (**d**) GSH concentration; (**e**) SOD activity; and (**f**) 8-OHdG concentration. Different lowercase letters indicate significant differences (*p* < 0.05) between groups in the plots.

**Figure 7 toxics-11-00877-f007:**
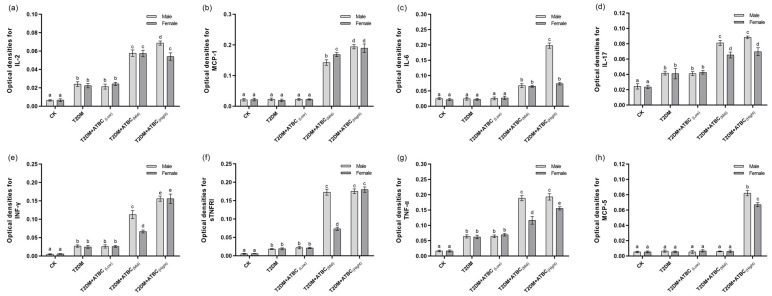
Cytokine level changes in mouse brains after ATBC exposure. (**a**) IL-2; (**b**) MCP-1; (**c**) IL-6; (**d**) IL-17; (**e**) INF-γ; (**f**) sTNFRI; (**g**) TNF-α; and (**h**) MCP-5. Different lowercase letters indicate significant differences (*p* < 0.05) between groups in the plots.

**Figure 8 toxics-11-00877-f008:**
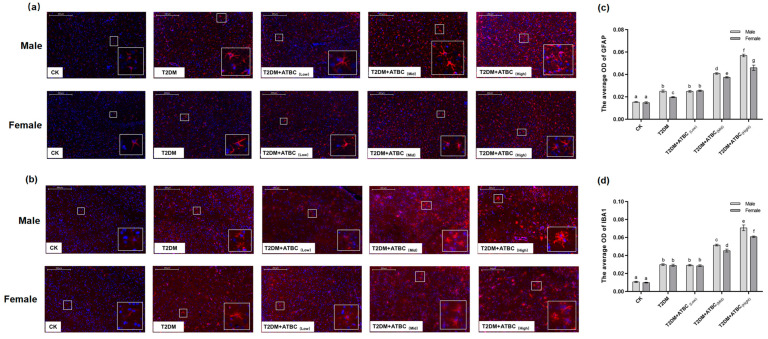
GFAP and IBA1 expression in mouse brains after ATBC exposure. (**a**) GFAP (Red part is GFAP expression); (**b**) IBA1 (Red part is GFAP expression). (**c**) The average OD (Red) of GFAP expression; (**d**) the average OD (Red) of IBA1 expression. (bar = 200 μm). Different lowercase letters indicate significant differences (*p* < 0.05) between groups in the plots.

**Figure 9 toxics-11-00877-f009:**
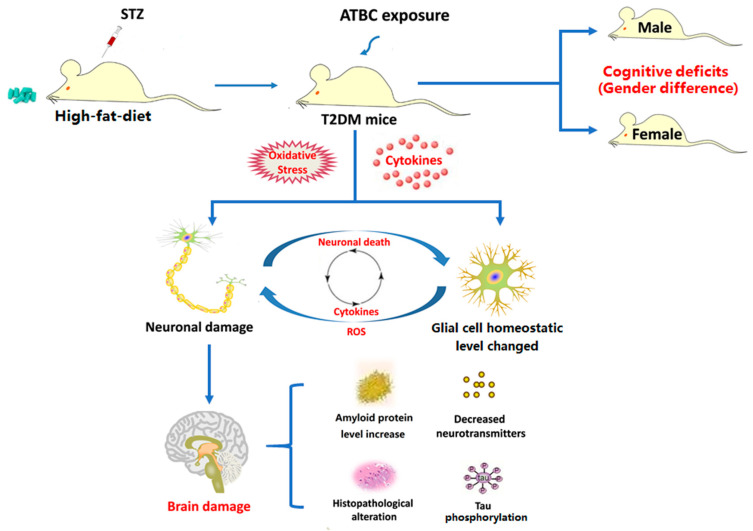
Potential mechanism of ATBC exposure-induced damage in T2DM mouse brain. Figure 9 was originally made by Xudong Liu.

**Table 1 toxics-11-00877-t001:** Mouse grouping and treatment.

Group	Treatment	Animal Number
CK (male/female)	ND + Normal saline	10
T2DM (male/female)	FD + Normal saline	10
T2DM + ATBC_(Low)_ (male/female)	FD + 2 mg/kg/day ATBC	10
T2DM + ATBC_(Mid)_ (male/female)	FD + 20 mg/kg/day ATBC	10
T2DM + ATBC_(High)_ (male/female)	FD + 200 mg/kg/day ATBC	10

## Data Availability

Not applicable.

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
