# Peer review of "Acetyl Tributyl Citrate Exposure at Seemingly Safe Concentrations Induces Adverse Effects in Different Genders of Type 2 Diabetes Mice, Especially Brain Tissue"

_toxics, 2023, doi:10.3390/toxics11100877_

Round 1
Reviewer 1 Report
There are two fundamental flaws in this manuscript which caused me to pause my review. The flaws were so significant that the manuscript cannot be published.
The authors should be measuring/graphing/analyzing how long it takes to find the platform across trails. Appropriate statistics should accompany all analysis. As it is, the water maze results have been analyzed subjectively.
As for GFAP and Iba1, those images aren't good enough to do appropriate analysis. For GFAP they should have measured area covered by GFAP or intensity which they said they did, but they don't show it. There's too much noise in these images to do intensity though. For Iba1 they should do Sholl analysis for morphology and they could also do an intensity measure although not ideal. They should've also done density of Iba1 expressing cells. Also they shouldn't say microglia "activation" anymore. https://pubmed.ncbi.nlm.nih.gov/36327895/

Author Response
1. The authors should be measuring/graphing/analyzing how long it takes to find the platform across trails. Appropriate statistics should accompany all analysis. As it is, the water maze results have been analyzed subjectively.
Response: Thanks for your good suggestion. We have performed an analysis of the time it takes mice to find the platform, the escape latency. Our analysis of this part is based on objective data, without subjective analysis, which may cause some misunderstandings about the description language of the results. We have made corresponding modifications in the paper to make the expression of the results more objective. As for the presentation of statistics, we mainly compared the trend and degree of change in different indicators among the groups. We believe that the statistical figures can clearly reflect the trend and degree of data change, and there is no need to repeat present statistics. In addition, there are too many indicators detected in this study, and animal groups of different genders are involved, so presenting the statistics of each group is not conducive to the fluency of the manuscript.
2. As for GFAP and Iba1, those images aren't good enough to do appropriate analysis. For GFAP they should have measured area covered by GFAP or intensity which they said they did, but they don't show it. There's too much noise in these images to do intensity though.
Response: Thanks for your good suggestion. During the study, we obtained the results after statistical analysis of the optical density of immunofluorescence expression of GFAP and Iba1. We did not present the statistical results in the previous manuscript, and now we have added the statistical result in the manuscript.
3. For Iba1 they should do Sholl analysis for morphology and they could also do an intensity measure although not ideal. They should've also done density of Iba1 expressing cells. Also they shouldn't say microglia "activation" anymore.
Response: Thanks for your good suggestion. We have supplemented the optical density analysis of Iba1 and modified the relevant expression of "activation" based on your recommended references.
The reviewer’s comments in the article annotations
1. Reword “high-safety”
Response: Thanks for your good suggestion. We have deleted it in the article.
2. What was the purpose of injecting STZ? How/why does the STZ prove that T2DM mice are a good model? Explanation is needed.
Response: Thanks for your good suggestion. Low-dose STZ injection combined with high-fat diet induced T2DM model is a classical and commonly used model construction method. Relevant researchers are very familiar with this model construction method, and we also cited relevant references. Therefore, we think that the detailed model construction method does not need to be introduced in manuscript. Simply, STZ as a widely used chemical inducer in animal models of diabetes, has destructive effects on pancreatic β cells in some animals. One-time large dose injection can cause direct damage of pancreatic β cells and induce rapid onset diabetes. Multiple low-dose injections can induce delayed onset diabetes, reduce animal mortality, and effectively simulate the course and pathogenesis of diabetes. STZ combined with a high-fat diet can induce a satisfactory T2DM model.
3. What does “NO” represent?
Response: Thanks for your good suggestion. We have revised the table.
4. How were the animals sacrificed?
Response: Thanks for your good suggestion. The mice were sacrificed by cervical dislocation. We have supplemented the description in manuscript.
5. References and kit names are needed.
Response: Thanks for your good suggestion. We have introduced the manufacturer of the kit and added the product number. The relevant use method can be carried out according to the product instructions, we do not have to describe the methodology in detail if our work in accordance with the manufacturer's recommendations.
6. References and kit names and more details are needed.
Response: Thanks for your good suggestion. We have introduced the manufacturer of the kit and added the product number. The relevant use method can be carried out according to the product instructions, we do not have to describe the methodology in detail if our work in accordance with the manufacturer's recommendations.
7. “rout” should be “route”.
Response: Thanks for your good suggestion. We have revised in the manuscript.
8. All of the routes look irregular to me. Unless you employ a statistical analysis of the route, I do not understand how you can make a statement on irregularity.
Response: Thanks for your good suggestion. There are some problems in the expression of our results. Through the heat map of mouse swimming routes, we can find that the relatively concentrated areas (red areas) of each group mice are different. With the increase of ATBC exposure concentration, the concentrated routes of mice in the SW quadrant gradually decrease. These results indicated that the spatial memory ability of mice to the location of the platform decreased, and the cognitive ability was affected. We have revised the expression of the results in the manuscript and added the quantitative analysis of the red region in the heat map of the swimming routes.
9. Again, without some type of statistical measurement, I don't agree that you can make statements about differences associated with toxicant exposure.
Response: Thanks for your good suggestion. Observation of the section can most directly detect changes in tissue structure. We believe that the sections provided by us have been able to intuitively see the changes of liver and pancreatic tissues after ATBC exposure, and the most important changes are the cytoplasmic sparsity and vacuolation of liver cells and the atrophy of islets. In order to better illustrate the reliability of the results, we conducted quantitative analysis of the optical density of the vacuolar region of liver cells and the the total area of islets, and the results have been supplemented in the manuscript.
10. How were these statistically compared?
Response: Thanks for your good suggestion. We believe that we can intuitively find the changes of pyramidal cells after ATBC exposure by observing the tissue sections. In order to better explain the results, we supplemented the quantitative analysis of the degree of cellular vacuolation and the results have been supplemented in the manuscript.
11. “Plenty” is not a statistical value. I don't agree that there is a difference.
Response: Thanks for your good suggestion. We supplemented the optical density analysis to explain the color depth of Nissl staining, and thus the amount of Nissl substance.
12. Provide alpha levels in all statical figures, please.
Response: Thanks for your good suggestion. We have added the statistics in the manuscript.
13. If the differences were not statistically different, there is no point in mentioning slight insignificant differences.
Response: Thanks for your good suggestion. We have deleted it in the manuscript.
14. Compare the medium and high concentrations to environmentally-relevant concentrations.
Response: Thanks for your good suggestion. We have supplemented the contents in discussion and conclusions parts of the manuscript, and supplemented the relevant references.
We have supplemented the contents in discussion and conclusions parts of the manuscript, and supplemented the relevant references.
15. I don't agree that the MWM showed significant differences.
Response: Thanks for your good suggestion. We have made changes in the manuscript to illustrate the reliability of the MWM results.
16. I am not convinced that there is a true statistical difference.
Response: Thanks for your good suggestion. We have made changes in the manuscript to illustrate the reliability of the results.
17. This was not statistically tested.
Response: Thanks for your good suggestion. We supplemented the optical density analysis to explain the color depth of Nissl staining, and thus the amount of Nissl substance.

Reviewer 2 Report
he manuscript by Zhang and colleagues reported the effects of ATBC exposure on T2DM mice, especially on the brain, by using a range of safe concentrations. The results showed that at seemingly safe concentrations, ATBC exposure exacerbated the disorder in glucose and lipid metabolism in T2DM mice and induced adverse effects on the T2DM mouse brain. Even if the topic is fascinating and interesting, some relevant concerns should be addressed:
- how can the author explain that mice fed with decreased their body weight since they showed increase in food intake, TC, TG and LDL levels - The FBG levels (lines 200 page 6) should be compared and discussed between T2DM and T2DM with ATBC - At brain levels T2DM did not show any significant difference, thus ATBC induce brain damage independently from diabetes - To evaluate pancreatic and liver damage a quantitative analysis should be performed (islet number and dimension, or hepatocyte ballooning etc) - Based on pancreas histological analysis in female there are not detectable islet: this means that insulin levels should be lower than in mice: why the author observe the opposite results? - In figure 5 should be reported all western blot; moreover, tau and GAPDH are referred to? - ROS levels in T2DM groups increased and GSH levels decreased: however, brain histological analysis did not reveal any minimal sign of alteration: how can the author explain these findings? - Similar results were observed for inflammatory cytokines: please explain - please correct IBA1 or Iba1 - Since in T2DM brain morphology is not altered, has the author tested whether ATBC alone alters brain function in control mice without T2DM?Author Response
1. How can the author explain that mice fed with decreased their body weight since they showed increase in food intake, TC, TG and LDL levels?
Response: Thanks for your good suggestion. Patients with increased diet and weight loss is a typical symptom of diabetes. Patients with elevated blood sugar and large amounts of glucose excreted from the body will reduce energy sources, which will increase lipolysis and lead to weight loss. Abnormal lipid metabolism is one of the major complications of diabetes, and many patients show elevated TC, TG and LDL. At present, there are many mechanisms may be involved, such as LDL receptor (LDLR) gene mutation mediated uptake. The relevant indicators in this study were mainly tested to determine whether the mice had diabetes-related symptoms and whether the symptoms would change under ATBC exposure.
2. The FBG levels (lines 200 page 6) should be compared and discussed between T2DM and T2DM with ATBC.
Response: Thanks for your good suggestion. We have supplemented the description and discussion of the relevant results in the manuscript.
3. At brain levels T2DM did not show any significant difference, thus ATBC induce brain damage independently from diabetes.
Response: Thanks for your good suggestion. We disagree with this view. Brain damage, though a common complication of diabetes, arises in response to certain conditions during the course of diabetes. The ATBC exposure concentration used in this study is widely considered to be relatively safe for normal population, but it is likely to induce brain injury in diabetes population. The premise for this kind of damage is diabetes, so we believe that this is diabetes-based brain injury and is not independent. It also indicates that the seemingly safe exposure concentration of ATBC is not safe for diabetic patients, which is the main point of this paper.
4. To evaluate pancreatic and liver damage a quantitative analysis should be performed (islet number and dimension, or hepatocyte ballooning etc).
Response: Thanks for your good suggestion. We have supplemented the description and discussion of the relevant results in the manuscript.
5. Based on pancreas histological analysis in female there are not detectable islet: this means that insulin levels should be lower than in mice: why the author observe the opposite results?
Response: Thanks for your good suggestion. According to pancreatic histopathology observation, we found that the islets of female and male T2DM mice did not disappear after exposure to high concentration of ATBC, and the islets were still visible, but the edges and boundaries of the islets were unclear, and the morphological structure was changed. The results of the insulin test also showed a decrease in insulin levels compared to the other groups, and the two results were not in conflict.
6. In figure 5 should be reported all western blot; moreover, tau and GAPDH are referred to.
Response: Thanks for your good suggestion. TAU: Total Tau protein, GAPDH: Intrinsic reference protein. We have supplemented the description and discussion of the relevant results in the manuscript.
7. ROS levels in T2DM groups increased and GSH levels decreased: however, brain histological analysis did not reveal any minimal sign of alteration: how can the author explain these findings? - Similar results were observed for inflammatory cytokines: please explain.
Response: Thanks for your good suggestion. We think this is a normal result. Although the brain of T2DM mice has certain index abnormalities at the molecular level, the abnormalities have not reached the level of causing brain tissue structural abnormalities. Only under high levels of oxidative stress and inflammation, abnormal levels exceed the compensatory level of the body and continue to accumulate, which will cause obvious damage to the cell structure, and the structural damage is also the result of abnormal long-term accumulation at the molecular level. This can also explain that not all T2DM patients will be complicated with brain injury, and when affected by some environmental factors, inducing sustained oxidative stress and inflammatory response can be regarded as one of the mechanisms to induce T2DM complicated with brain injury.
8. Please correct IBA1 or Iba1.
Response: Thanks for your good suggestion. We use IBA1 as the correct expression in the whole paper, and have modified it in the manuscript.
9. Since in T2DM brain morphology is not altered, has the author tested whether ATBC alone alters brain function in control mice without T2DM?
Response: Thanks for your good suggestion. First, according to the reference, the ATBC exposure concentrations we used in this paper are safe for normal mice. At the same time, we have done previous studies to prove that ATBC does not cause functional impairment of brain tissue in normal mice at moderate concentrations as described in this paper (academic papers have been published). In this study, ATBC exposure at moderate concentration can significantly cause brain tissue abnormalities and functional abnormalities in T2DM mice. This suggests that exposure concentrations that seem safe may not be safe for sensitive groups, which is the main point of this paper. We have supplemented the relevant discussion and literature citations in the manuscript.

Reviewer 3 Report
Review of the manuscript entitled: Acetyl tributyl citrate exposure at seemingly safe concentrations induces adverse effects in different genders of type 2 diabetes mice, especially brain tissue. In my opinion, the manuscript is very interesting. Moreover, it raises a very important topic! The abstract and introduction are prepared correctly.
However, some questions arise:
- is ATBC detected in the brain?
- if you use commercial kits, provide the catalog number, this enables research replication. Moreover, you do not have to describe the methodology in detail if you work in accordance with the manufacturer's recommendations.
- in the description of table 1, add the number of animals in the group
- Fig. 1 is illegible. Is it possible to enlarge the font? or bold? Similar Fig 5, 6 and 7.
- Is there anything known about the effect of ATBC on the PPAR gamma receptor? it is crucial in the development of AD and T2DM. In addition, many EDCs affect PPARgamma
Author Response
1. Is ATBC detected in the brain?
Response: Thanks for your good suggestion. We did not measure the amount of ATBC in the mouse brain, because the final poison of many substances is not necessarily the substance itself, but its metabolites, or endogenous substances stimulated during metabolism. It has been reported that ATBC exposure can affect animal behavior and has potential neurotoxicity. Research presented recently at Discover BMB, the annual meeting of the American Society for Biochemistry and Molecular Biology, in Seattle, showed that neuroblastoma cells exposed to ATBC increased the expression of two genes associated with cellular stress (Nrf2 and p53), as well as the production of enzymes associated with cellular aging. ATBC may interfere with glial cells' ability to regenerate, thereby reducing their ability to protect nerve cells, leading to neurodegeneration and accelerated aging. We have reason to believe that ATBC body or metabolite have the potential to cause neurotoxicity and this effect may be more sensitive to special populations. Therefore, we did not detect the presence or absence of ATBC in mouse brains, but focused on the effects of different doses of ATBC exposure on mice, especially brain tissue. Of course, whether ATBC produces neurotoxicity through itself, in the form of metabolites, or by endogenous substances stimulated during metabolism is a question that is well worth exploring after determining its toxic effects. The results of this study have shown that oxygen free radicals and inflammatory factors in brain tissue can be stimulated after ATBC exposure, and relevant studies need to be further explored in the later stage. We have added the possible future work to the end of this article.
2. If you use commercial kits, provide the catalog number, this enables research replication. Moreover, you do not have to describe the methodology in detail if you work in accordance with the manufacturer's recommendations.
Response: Thanks for your good suggestion. We have supplemented the numbers of the kits used in the manuscript and simplified the relevant methods.
3. In the description of table 1, add the number of animals in the group.
Response: Thanks for your good suggestion. We have revised the table to add the number of animals in each group.
4. Fig. 1 is illegible. Is it possible to enlarge the font? or bold? Similar Fig 5, 6 and 7.
Response: Thanks for your good suggestion. We have reformatted the figures and enlarged it to make it easier to observe.
5. Is there anything known about the effect of ATBC on the PPAR gamma receptor? it is crucial in the development of AD and T2DM. In addition, many EDCs affect PPARgamma.
Response: Thanks for your good suggestion. There is no ATBC report on PPAR-γ. In this study, we explored the potential mechanism of ATBC on brain tissue of T2DM mice from the perspective of oxidative damage, neuroinflammation and glial cell activation. PPAR-γ can regulate inflammatory response through JAK-STAT, AP-1, NF-κB, NFAT, etc. This may be the direction that our next research can explore. We have added the possible follow-up work to the end of the manuscript.

Round 2
Reviewer 1 Report
The authors seemed to have addressed my concerns.
The authors seemed to have addressed my concerns.
Author Response
Thanks for your careful review of the article again. We have checked and modified the article again to make it more perfect.
Reviewer 2 Report
The authors answered all my concerns, significantly improving the manuscript. However, a minor concern should be addressed before considering the manuscript for publication. - Regarding my previous concern: At brain levels T2DM did not show any significant difference, thus ATBC induces brain damage independently from diabetes, the authors answered as follow: "We disagree with this view. Brain damage, though a common complication of diabetes, arises in response to certain conditions during the course of diabetes. The ATBC exposure concentration used in this study is widely considered to be relatively safe for normal population, but it is likely to induce brain injury in diabetes population. The premise for this kind of damage is diabetes, so we believe that this is diabetes-based brain injury and is not independent. It also indicates that the seemingly safe exposure concentration of ATBC is not safe for diabetic patients, which is the main point of this paper". I agree with the authors whether they can demonstrate that ATBC alone in control non diabetic group (in the same experimental condition proposed) did not affect brain function, otherwise it is a speculation but not a result.
Author Response
Thanks for your good suggestion. We have noted this issue in previous studies and found that 20 mg/kg/day did not affect brain function in normal mice, while in this study the exposure concentration significantly affected brain function in diabetic mice. Our previous research has been published in academic papers (reference [16], [17]), and in the “Introduction” part of R1, we have supplemented relevant explanations and cited the literatures (reference [16], [17]). Perhaps in the “Conclusions” part, we did not highlight this issue, so in R2 we added relevant content to show that the seemingly safe ATBC expose-induced brain damage in diabetic mice does not exist independently.